# ON REPRESENTATION LEARNING UNDER CLASS IMBALANCE

## ABSTRACT

Unlike carefully curated academic benchmarks, real-world datasets are often highly class-imbalanced, involving training and test sets which contain few examples from certain minority classes. While there is a common understanding that neural network generalization is negatively impacted by imbalance, the source of this problem and its resolution are unclear. Through extensive empirical investigation, we study foundational learning behaviors for various models such as neural networks, gradient-boosted decision trees, and SVMs across a range of domains and find that (1) contrary to conventional wisdom, re-balancing the training set to include a higher proportion of minority samples degrades performance on imbalanced test sets; (2) minority samples are hard to fit, yet algorithms which fit them, such as oversampling, do not improve generalization. Motivated by the observation that re-balancing class-imbalanced training data is ineffective, we show that several existing techniques for improving representation learning are effective in this setting: (3) self-supervised pre-training is insensitive to imbalance and can be used for feature learning before fine-tuning on labels; (4) Bayesian inference is effective because neural networks are especially underspecified under class imbalance; (5) flatness-seeking regularization pulls decision boundaries away from minority samples, especially when we seek minima that are particularly flat on the minority samples' loss.

## 1 INTRODUCTION

In real-world data collection scenarios, some events are common while others are exceedingly rare. For example, only a miniscule proportion of credit card transactions are fraudulent, and most cancer screenings come back negative. As a result of this property, machine learning systems are routinely trained and deployed on class-imbalanced data where relatively few samples are associated with certain *minority classes*, while *majority classes* dominate the datasets. Nonetheless, the vast majority of works exclusively consider class balanced benchmarks (LeCun, 1998; Krizhevsky, 2009; Deng et al., 2009), including both foundational literature which seeks to understand how and why machine learning algorithms operate as well as applied methodological literature.

In this work, we conduct an exploration, on various machine learning approaches including neural networks, gradient-boosted decision trees, and SVMs, of what makes learning under class imbalance so difficult and the associated implications for best practices in such scenarios. Many of the widely referenced methods for remedying class-imbalance problems rely on modifying how the training data is sampled, such as oversampling or SMOTE (Chawla et al., 2002) and have been shown to be ineffective for neural networks (Buda et al., 2018).

A common assumption underpinning these sampling methods is that learning under class imbalance is pathological, perhaps even defaulting to predicting only the majority class on all inputs when imbalance is sufficiently severe, so we must intervene by simulating balanced training. An effect of sampling from minority classes disproportionately more often is that more signal from those classes is injected into model updates, helping to fit the otherwise rarely seen minority samples. To tease out exactly why oversampling is ineffective, we begin by studying the relationship between imbalances seen at train and test time, and we investigate whether poor generalization under class-imbalance can really be explained by failures of optimization. We find that while minority samples are hard to fit, this optimization phenomenon has little explanatory power regarding generalization as fitting them

does not affect performance. Additionally, we find that on the one hand, re-balancing training data to include more minority samples can negatively impact generalization under imbalanced testing, but on the other hand, gathering more majority samples to increase the size of the dataset degrades generalization as well.

Following our investigation into the role of dataset imbalances in generalization, we show why self-supervised learning (SSL), Bayesian inference, and flatness-seeking regularizers are particularly well-suited for deep learning in class-imbalanced settings. Self-supervised learning algorithms are less sensitive to the proportion of samples in various classes as they do not make use of label information, so we can learn better feature representations via SSL before fine-tuning even on the same data but with labels. Previous works have found that the number of high singular values of the Hessian is related to the number of classes (Sagun et al., 2017; Papyan, 2020), but these works were conducted strictly on balanced data. In examining such properties on imbalanced datasets, we observe that neural networks trained in such settings are significantly more underdetermined by the data. In cases where many parameter settings, and induced functions, are compatible with the data, Bayesian Neural Networks (BNNs) can represent our uncertainty for improved accuracy (Wilson & Izmailov, 2020; Shwartz-Ziv et al., 2022), and we find that this capability of BNNs is especially advantageous in the class-imbalanced regime. Finally, whereas neural network decision boundaries tend to hug minority samples in order to expand the margins from majority data points which occur more frequently in training data, we can counteract this behavior with Sharpness-Aware Minimization (SAM) (Foret et al., 2020), and we can further improve margins with respect to minority samples by increasing flatness on their corresponding loss functions. In summary, our work questions the motivation of orthodox sampling methods and proposes new directions by which improved representation learning can benefit classifiers in class-imbalanced settings.

## 2 RELATED WORKS

A long line of research has been conducted on imbalanced classification. There are several general approaches to address this problem: (1) **Re-sampling the data -** In early ensemble learning studies, boosting and bagging algorithms were adjusted to take account of imbalanced data by re-sampling. Traditionally, re-sampling involves oversampling minority class samples by simply copying them (Guo & Viktor, 2004; Chawla et al., 2002; Han et al., 2005), or undersampling majority classes by removing samples (Drummond et al., 2003; Hu et al., 2020; Ando & Huang, 2017; Buda et al., 2018), so that minority and majority class samples appear equally frequently in the training process. (2) **Loss re-weighting**: Loss re-weighting assigns different weights to majority and minority classes, thus reducing optimization difficulty under class imbalance (Cui et al., 2019; Huang et al., 2019a). For instance, one may scale the loss by inverse class frequency He & Garcia (2009) or re-weight it using the effective number of samples Cui et al. (2019). As an alternative approach, one may focus on hard examples by down-weighing the loss of well-classified examples (Lin et al., 2017) or dynamically re-scaling the cross-entropy loss based on the difficulty of classifying a sample (Ryou et al., 2019). Bertsimas et al. (2018) propose to encourage larger margins for rare classes, while Goh & Sim (2010) learn robust features for classify minority classes using class-uncertainty information which approximates Bayesian methods. 3) **Two-stage fine-tuning and meta-learning approaches:** Two-stage methods separate the training process into representation learning and classifier learning (Liu et al., 2019; Ouyang et al., 2016; Kang et al., 2019; Bansal et al., 2021). In the first stage, the data is unmodified, and no re-sampling or re-weighting is used to train good representations. In the second stage, the classifier is balanced by freezing the backbone and fine-tuning the last layers with re-sampling techniques or by learning to debias the confidence. These methods assume that the bias towards majority classes exists only in the classifier layer or that tweaking the classifier layer can correct the underlying biases.

Several works have also inspected representations learned under class imbalance. Kang et al. (2019) find that representations learned on class-imbalanced training data via supervised learning perform better when the linear head is fine-tuned on balanced samples. Yang & Xu (2020) instead examine the effect of self- and semi-supervised training on imbalanced data and conclude that imbalanced labels are significantly more useful when accompanied by auxiliary data for semi-supervised learning. Kotar et al. (2021); Yang & Xu (2020); Liu et al. (2021) make the observation that self-supervised pre-training is insensitive to imbalance in the upstream training data. These works study SSL pre-training for the purpose of transfer learning, sometimes using linear probes to evaluate the quality of

representations. Inspired by their observations, we find that SSL pre-training and then fine-tuning on the same exact class-imbalanced dataset can significantly improve generalization.

## 3 EXPERIMENTAL SETUP

**Class-imbalance ratio.** Let $r$ denote the ratio between the number of samples in the rarest class to the number of samples in the most frequent class. In the case where $r = 1$, the dataset is fully balanced, whereas $r = 0.1$ indicates that there are 10 times as many samples in the majority class as in the minority class. In this paper, we will construct and investigate both *train* and *test* sets with varying imbalance ratios. It is important to note that if the train dataset ratio is fixed and the test ratio is varied, then we may train a single model but evaluate it on a variety of test sets. On the other hand, if the test ratio is fixed and the training ratio is varied, we must train many models. We will vary our setup along both these axes.

**Datasets.** For experiments with neural networks, we use CIFAR-10 (Krizhevsky, 2009) as well as a binary variant in which we simply use two of the CIFAR-10 classes. For tabular data, we use the Adult dataset and Forest Cover dataset from the UCI Machine Learning Repository (Dua & Graff, 2017). We adopt the imbalanced datasets of Liu et al. (2019) which imbalance classes via exponential distribution, closely reflecting natural long-tailed class distributions.

**Models.** We use ResNet-34 (He et al., 2016) on CIFAR-10. We use XGBoost (Chen & Guestrin, 2016) and SVM on tabular datasets. For supervised pre-training, we follow the standard protocol of (Kang et al., 2020). For self-supervised pre-training, we use SimCLR (Chen et al., 2020) and fine-tune as in (Kotar et al., 2021). For each evaluation in our experiments, we run five seeds and report the mean along with one standard error. Appendix A.1 contains additional details.

## 4 THE ROLE OF IMBALANCED DATA IN GENERALIZATION AND OPTIMIZATION

In this section, we first explore the impact of training set imbalances on generalization across various imbalanced testing scenarios, finding that re-balancing training data actually harms accuracy under imbalanced testing, across multiple ML models. We then find that severe class imbalance makes fitting minority samples difficult, but solving this problem is irrelevant to generalization in neural networks. Re-balancing the training data and fitting minority samples are both primary effects of oversampling, so these observations will inspire us to look past sampling algorithms in our subsequent search for improved training methods for neural networks.

### 4.1 THE RELATIONSHIP BETWEEN TRAIN-TIME AND TEST-TIME IMBALANCE

The literature on training routines for class imbalance in machine learning is filled with methods designed for scenarios in which training data is highly imbalanced, but testing data is balanced. However, in industrial applications of machine learning, data encountered during deployment is typically also imbalanced. Therefore, we disentangle training and testing balances and investigate how sensitive models are to discrepancies between the two. This study may be particularly important if one is considering collecting training data for a downstream application. Should we gather training data with the same balance as we anticipate during testing? If the data we encounter during deployment is more or less balanced than the training data we gathered, how worried should we be?

A priori, the optimal train set balance is unclear. On the one hand, one might want to train on the same distribution as they anticipate testing on. On the other hand, previous studies have suggested that imbalanced training data impedes representation learning and have suggested sampling procedures for simulating more balanced training (Gosain & Sardana, 2017). To answer this question, we train on datasets with a wide range of imbalance ratios and evaluate each trained model on a variety of testing ratios. We illustrate three scenarios in Figure 1: (1) identical training and testing ratios, (2) balanced training, and (3) the training ratio with the lowest test error (optimal training ratio). We see that training on data with the same imbalance as the testing data is typically superior to training on balanced data, and the two strategies only approach equal performance when the testing data

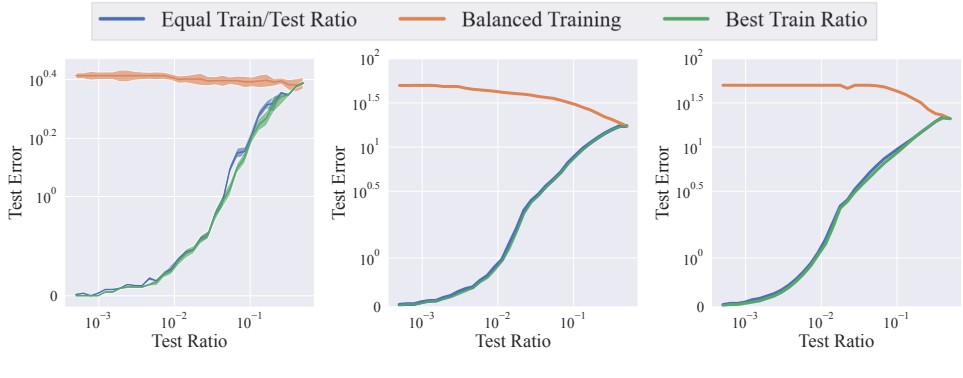

(a) CIFAR-10 Dataset      (b) XGBoost on Adult Dataset     (c) SVM on Forest Cover Dataset

Figure 1: **Training on imbalanced data is optimal for imbalanced testing scenarios.** Test accuracy as a function of the test dataset ratio for different training setups. Error bars correspond to one standard error over 5 trials.

becomes balanced. Moreover, training on data with the same imbalance as the testing data is nearly optimal, so there is very little to gain by balancing the training set.

To determine the optimal training distribution, we plot the best train ratio versus the best test ratio in Figure 2. In this figure, if there is a perfect match between the training and testing distributions, we would expect points to lie on the diagonal. Indeed, the points are close to the diagonal, indicating that it is best to train with a very similar ratio to that of the test dataset, especially for highly imbalanced testing scenarios. Interestingly, when the optimal training distribution is not exactly the same as the testing distribution, it is usually very close to it and generally more balanced than the testing distribution. Even in cases where the best ratio for training is more balanced, there is minimal difference in test error between the best ratio for training and the test-equivalent ratio for training. This phenomena is consistent across different training methods (including oversampling training Drummond et al. (2003) and loss re-weighting Cui et al. (2019)). This observation calls into question oversampling routines which attempt to improve performance by re-balancing the training data.

> **Takeaway:** Training with a train set class imbalance similar to that of the test set is near optimal across the ML models and datasets we consider.

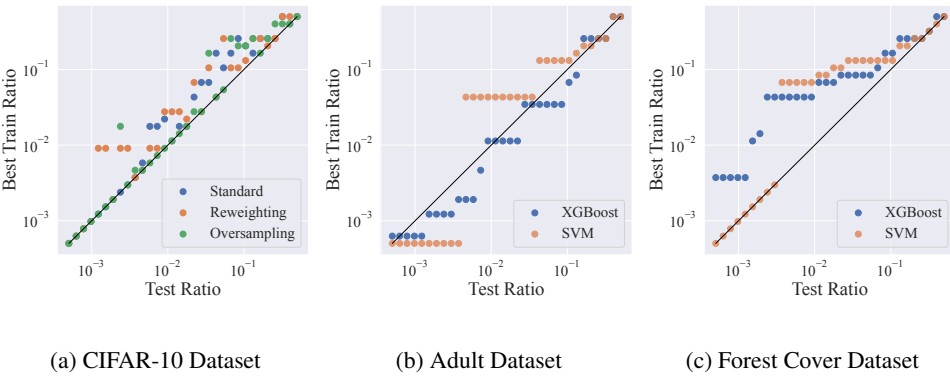

(a) CIFAR-10 Dataset       (b) Adult Dataset       (c) Forest Cover Dataset

Figure 2: **The optimal train dataset ratio is very close to the test dataset ratio.** Optimal train imbalance ratio as a function of test imbalance ratio for various datasets and models.

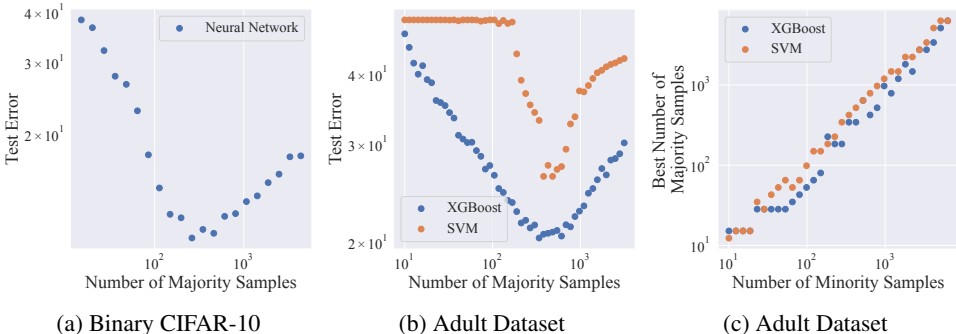

(a) Binary CIFAR-10    (b) Adult Dataset    (c) Adult Dataset

Figure 3: **The potentially destructive effects of adding majority class data**. In (a) and (b), we fix the number of minority samples to be 500 and vary the number of majority samples. In (c), we plot the number of majority samples that gives us the lowest test error against the number of minority samples. We see that increasing the number of majority samples degrades performance. Error reported on balanced test set.

### 4.2 WHEN MORE DATA DEGRADES PERFORMANCE

In the previous section, we fixed the total number of training samples and saw that models perform best when trained on a similar data distribution to their testing data. However, in practice, a practitioner likely will not have precise control over the data they collect. Will collecting additional samples always help performance? If not, practitioners must be cautious when collecting new data that its balance matches their testing data or else the additional data could result in even worse models. Instead of fixing the total number of samples and varying their class ratio, we now fix the number of samples from the minority class and vary the number of others.

In Figure 3, we see that increasing the number of samples from the majority class, both for neural networks (Figure 3a) and XGBoost (Figure 3b), initially boosts performance on a balanced test set. Nevertheless, in both cases, the performance reaches an optimum before the ever increasing imbalance in training data eventually degrades test accuracy. Thus, adding training data can be helpful, even without considering the balance of the additional data, but if we add enough samples, we need to be careful not to cause too sharp a mismatch between training and testing distributions. Notably, the optimal training set ratio is nearly balanced, matching the test set, even when we are allowed to gather extra samples from one class without having to forego samples from another.

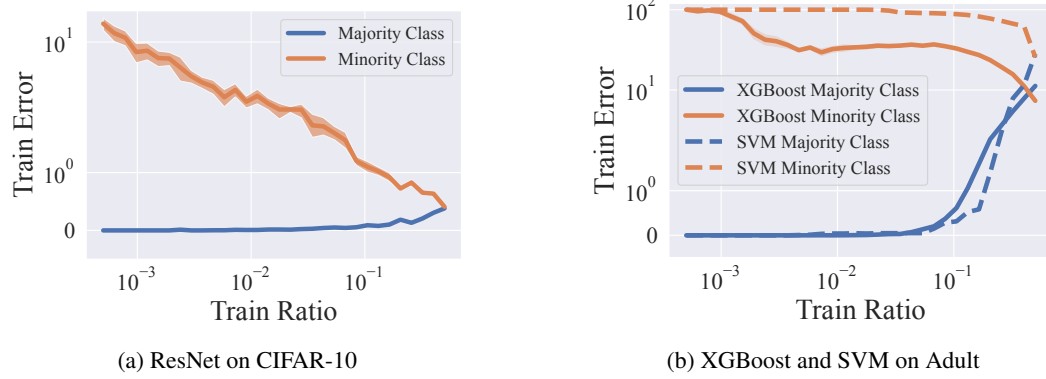

(a) ResNet on CIFAR-10    (b) XGBoost and SVM on Adult

Figure 4: **Standard training routines fail to fit minority samples.** Error bars correspond to one standard error over 5 trials

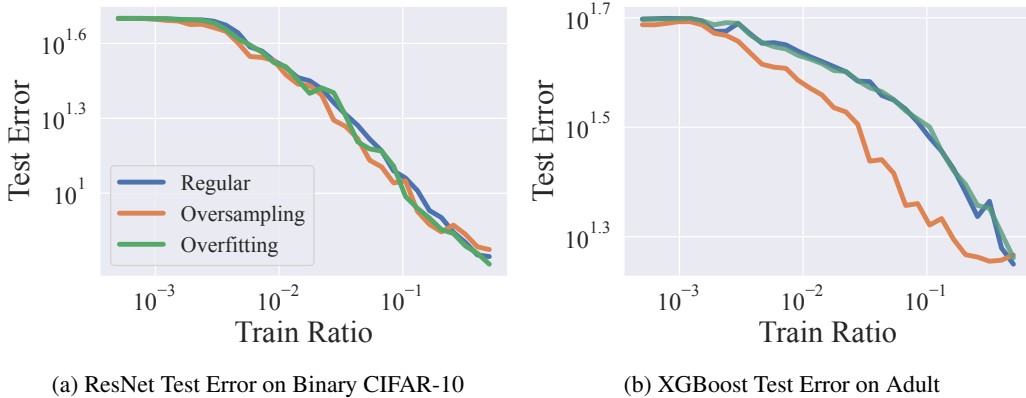

(a) ResNet Test Error on Binary CIFAR-10      (b) XGBoost Test Error on Adult

Figure 5: **Overfitting with a low learning rate or oversampling does not improve neural network generalization**. Oversampling is helpful for XGBoost. Error reported on a balanced test set.

### 4.3 MINORITY DATA IS HARD TO FIT, BUT FITTING IT DOES NOT HELP

We now investigate optimization on imbalanced data. Does the classifier correctly label minority training points? Our goal is to determine if low performance under class imbalance can be attributed to poor optimization. This question will inform our future exploration of methods for training on imbalanced data as it will tell us what underlying problem needs solving.

In imbalanced training, the vast majority of the gradient signal during training comes from majority class samples since they are seen much more often, making it hard to fit minority samples as shown in Figure 4. In order to re-balance imbalanced training data, practitioners employ various methods, including oversampling and undersampling. These methods sample minority data more often, thus boosting their contribution to the gradient and helping to fit these minority samples. We confirm in Appendix A.2 that such sampling methods indeed help to fit minority samples across models. But is the inability to fit minority samples a culprit responsible for worse test accuracy?

To answer the above question, we train our model using two additional methods: (1) oversampling the minority class, and (2) overfitting the training examples using a large number of epochs and a small learning rate. Both methods successfully fit minority and majority classes. Nonetheless, they fail to improve neural network test performance in contrast to XGBoost which benefits from oversampling, as seen in Figure 5. We include additional details and results for neural networks and XGBoost, as well as results on imbalanced test sets, in Appendix A.2. In light of this study, while minority samples are hard to fit, we will search elsewhere for improving neural network generalization in subsequent sections.

> **Takeaway:** Since minority samples are rarely seen in training, they are hard to fit. Nonetheless, strategies such as oversampling which fit these samples do not improve generalization.

## 5 COMBINING SELF-SUPERVISED LEARNING WITH SUPERVISED FINE-TUNING FOR ROBUST FEATURE LEARNING

Self-supervised learning (SSL) has recently achieved great success for representation learning in computer vision, NLP, and tabular data (Chen et al., 2020; Kenton & Toutanova, 2019; Somepalli et al., 2021). SSL pre-trained networks often exhibit even more transferable representations than their supervised counterparts (Grill et al., 2020). Liu et al. (2021) notes that self-supervised pre-training training strategies for transfer learning are more robust to upstream imbalance than supervised pre-training. However, many use-cases for deep learning are not accompanied by massive pre-training datasets. We thus propose a two-step procedure in which we first perform SSL pre-training and then supervised fine-tuning, all on the same imbalanced dataset. By first learning a

feature extractor via SSL, which is insensitive to class imbalance, we can improve the quality of features and as a result generalization to

We use SimCLR (Chen et al., 2020) for self-supervised pre-training and try two fine-tuning routines: (1) fine-tune all layers in an end-to-end fashion and (2) train only a fully-connected layer on top of the fixed SSL feature extractor. We train on CIFAR-10 using a wide range of class-imbalance ratios reporting accuracy on a balanced test set, and we compare to three baselines including standard supervised learning, oversampling, and SimCLR ImageNet pre-training. The latter baseline uses an extra one million training samples for pre-training, which may not be available for a particular data domain in practice, but will serve as an aspiration for representation learning. In Figure 6, we see that our two-step procedure which uses SSL pre-training on the small imbalanced CIFAR-10 dataset achieves almost as high performance as ImageNet pre-training and far superior performance to standard supervised learning and oversampling, even when the training set is relatively balanced. See Appendix Appendix A.2 for an identical experiment but where testing data has the same imbalance as the training data. We see a similar trend there, namely that SSL pre-training improves generalization significantly.

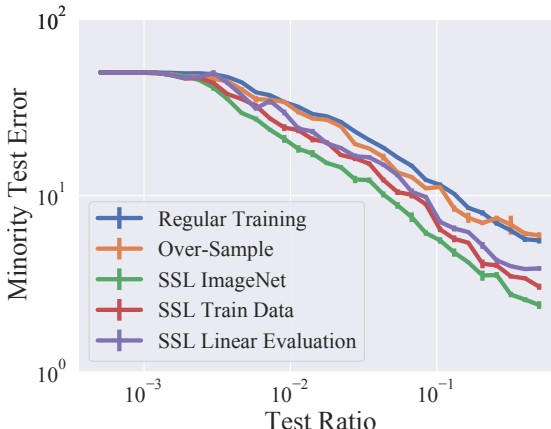

Figure 6: **SSL pre-training and end-to-end fine-tuning improves representation learning.** Test error reported on balanced test sets for neural networks on binary CIFAR-10. Error bars correspond to one standard error over 5 trials

> **Takeaway:** A two-step process of (a) SSL pre-training on the imbalanced training set and (b) supervised fine-tuning on the same data can boost accuracy across imbalance ratios.

## 6 UNDERSPECIFICATION AND BAYESIAN INFERENCE

Expressive models such as neural networks are often capable of representing numerous functions which are each compatible with the training data. Standard neural network training protocols only choose one of these solutions. However, we can better represent this uncertainty using an ensemble. Bayesian Neural Networks (BNNs) approach this problem by estimating the average prediction over all parameters weighted by their posterior probabilities conditioned on the training data, $p(y|x, \mathcal{D}) = \int p(y|x, w)p(w|\mathcal{D})dw$. This approach is particularly effective when the model is *underspecified* by the data, or in other words when there are many solutions that each achieve a high likelihood (Wilson & Izmailov, 2020). In this section, we see that underspecification is particularly strong on class-imbalanced data so that BNNs are especially useful in this setting.

## 7 FLATNESS-SEEKING REGULARIZATION PULLS DECISION BOUNDARIES AWAY FROM MINORITY SAMPLES

A number of works have found that the number of high singular values of the loss function Hessian is equal to the number of classes (Sagun et al., 2017; Papyan, 2020). Papyan (2020) notes that the theoretical underpinnings of this connection may depend on the training data being class balanced. We thus measure the 10 leading singular values of the Hessian on models trained on balanced or highly imbalanced CIFAR-10 data, where the loss function is evaluated on the models' respective training data. We observe in Figure 7a that imbalanced data leads to *lower* singular values. We also perturb the parameter vector in random directions in Figure 7b using filter-normalization (Huang et al., 2019b) to normalize distance measurements, and we see that the loss increases slightly slower

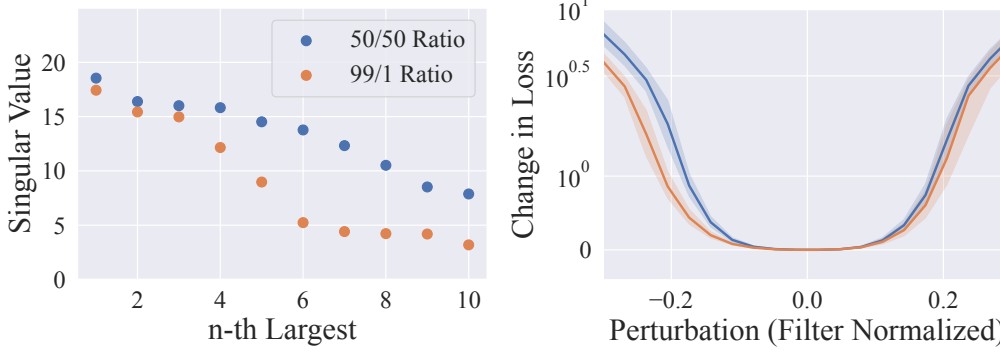

(a) Ten largest singular values of the Hessian.    (b) Loss minima are flatter under imbalanced data.

Figure 7: **Loss minima are flatter under imbalanced data**. Models trained on CIFAR-10.

on imbalanced data. Note that the volume of a basin scales with its radius raised to a power equal to the dimension of the space. As a neural network contains millions of parameters, a very small increase in basin radius will therefore lead to a massive increase in volume (Huang et al., 2019b). Flat minima have the property that their parameters can be perturbed without unfitting the data, indicating that there are many solutions, even locally, which are all consistent with the training samples. But which of these solutions should we choose? By using Bayesian inference methods, we need not commit to only a single one.

We now try two such Bayesian inference procedures. The Laplace approximation (Daxberger et al., 2021) fits a Gaussian distribution with diagonal covariance to the posterior so that lower curvature, which we observed above on imbalanced datasets, will induce higher variance. SGHMC (Chen et al., 2014) is a Markov chain Monte Carlo sampler which instead introduces noise to the training procedure so that parameters spend more time in places with higher probability under the posterior and samples checkpoints periodically. We see in Figure 8 that such methods confer especially large boosts in accuracy on class-imbalanced training data with virtually no additional training cost.

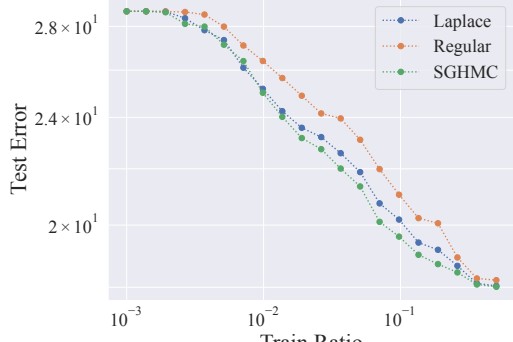

Figure 8: **Bayesian inference improves performance, especially on imbalanced training data.** Error reported on balanced CIFAR-10 test set.

> **Takeaway:** Under class imbalance, models are particularly underspecified by the data. We can reflect uncertainty over solutions with Bayesian inference, leading to increased accuracy.

Sharpness-Aware Minimization (SAM) (Foret et al., 2020) is an optimizer for finding flat minima of the loss function which often generalize better than those found by SGD. To this end, SAM entails an inner ascent step followed by a descent step which finds parameters such that an ascent step minimally increases the loss. Huang et al. (2019b) connect flat minima to wide margin decision boundaries. By plotting the decision boundaries of a small multi-layer perceptron on a toy 2D dataset in Figure 9, we see small margins surrounding minority class samples, and SAM expands these margins supporting the intuition of Huang et al. (2019b). Following this observation, we employ SAM on our CIFAR-10 setup and find that SAM especially improves generalization on class-imbalanced training data in Figure 10. In light of this result, we now further increase flatness specifically on minority class loss terms by increasing the ascent step size in SAM's inner loop. We see in Figure 10 that this adaptation can yield even greater performance boosts. See Liu et al. (2021) for an adaptation of SAM which, rather than increasing the size of ascent steps on minority sample loss, instead performs self-supervised pre-training where the ascent step size is related to the density of examples estimated via kernel density estimation.

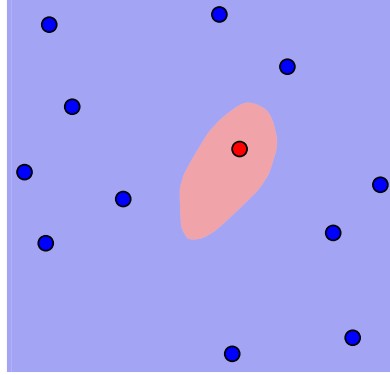 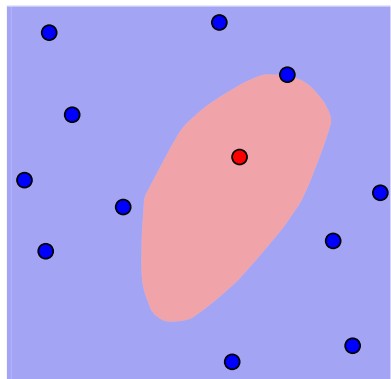

(a) Decision boundaries after regular training   (b) Decision boundaries after SAM training

Figure 9: **Flatness seeking regularization pulls decision boundaries away from minority samples.** Experiments conducted on toy 2D dataset paired with MLP architecture.

## 8 DISCUSSION

In this work, we first examined the effects of class-imbalanced data on optimization and generalization. The results of these experiments indicated that oversampling solves exactly the wrong problems, namely re-balancing the training data is not helpful, and fitting minority samples does not improve generalization. The observation that adding additional majority class samples actually hurts performance raises a critical open problem: how can we best harness extra majority class samples? Following the above study, we suggest three existing methods for improving performance on class-imbalanced data without ad-hoc interventions specific to imbalance: (1) since SSL is insensitive to imbalance, pre-training followed by end-to-end fine-tuning can result in enhanced representations; (2) due to amplified underspecification, Bayesian inference is particularly effective

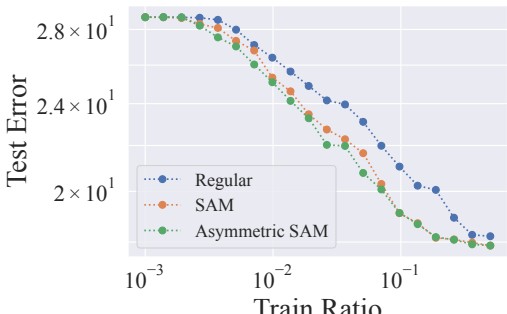

Figure 10: **Sharpness-Aware Minimization (SAM) is particularly effective on imbalanced training data.** "Asymmetric SAM" denotes the version where the inner loop ascent step size is increased for minority samples. Error reported on balanced CIFAR-10 test set.

on class-imbalanced problems; (3) sharpness-aware minimization can pull decision boundaries away from minority samples. Given the pervasiveness of imbalanced data in real-world applications, the success of these simple existing methods should redirect our attention away from re-sampling methods and towards better feature representations.

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

# A APPENDIX

## A.1 MODEL DETAILS

**XGBoost**: 'XGBClassifier' from XGBoost version 1.6.2

- 'n_estimators' = 100
- 'subsample' = 0.5
- 'eta' = 0.3
- 'max_depth' = 6

**SVM**: 'LinearSVM' from sklearn version 1.1.2

- 'dual' = false
- 'max_iter' = 1000

## A.2 SUPPLEMENTAL FIGURES

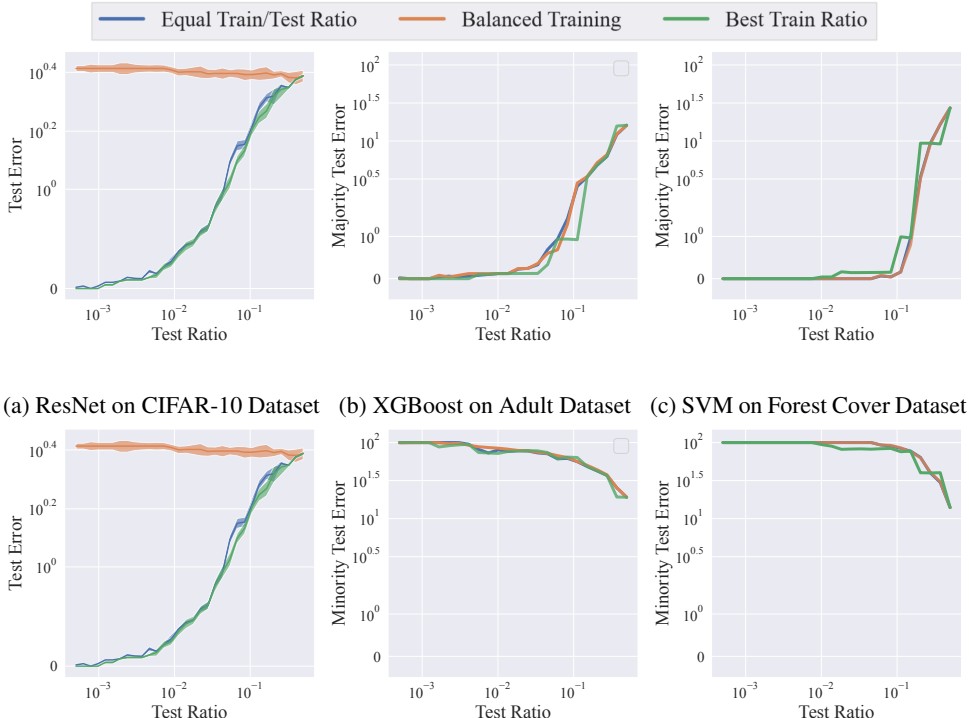

(a) ResNet on CIFAR-10 Dataset   (b) XGBoost on Adult Dataset   (c) SVM on Forest Cover Dataset

(d) ResNet on CIFAR-10 Dataset   (e) XGBoost on Adult Dataset   (f) SVM on Forest Cover Dataset

Figure 11: Test error split by majority and minority classes for balanced test sets. We see similar trends across all models and datasets.

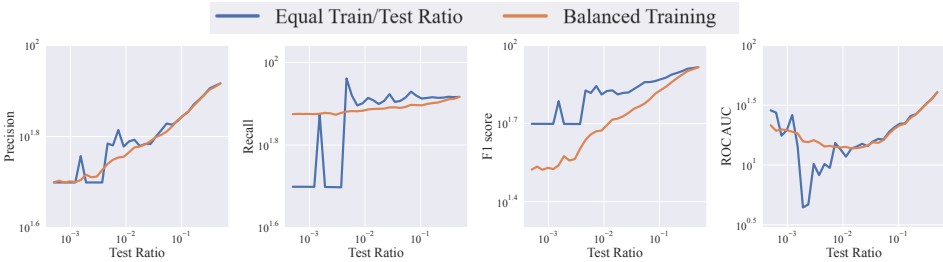

Figure 12: Additional metrics for XGBoost on Adult dataset

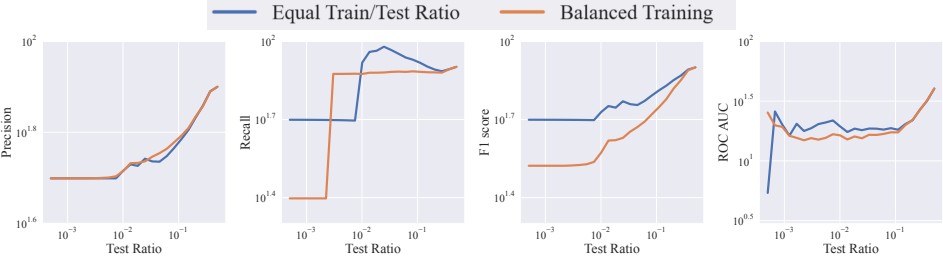

Figure 13: Additional metrics for SVM on Forest Cover dataset

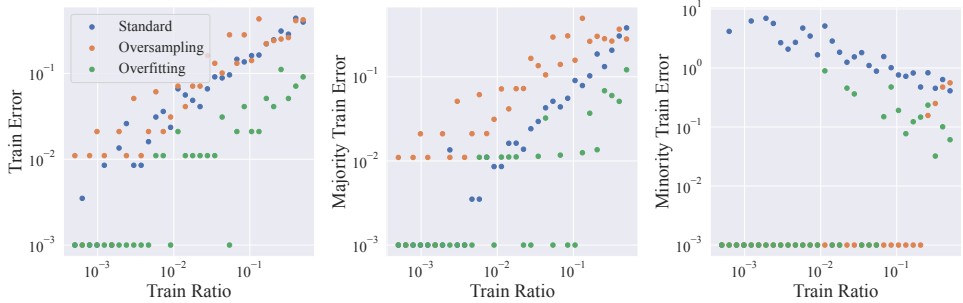

Figure 14: ResNet train error on CIFAR-10

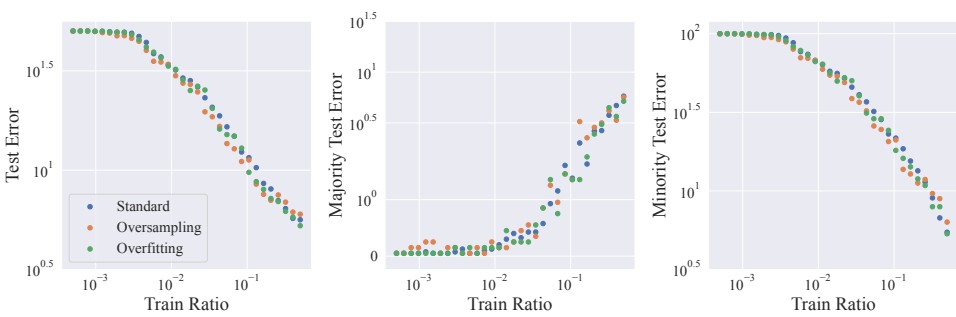

Figure 15: ResNet test error on imbalanced test sets from CIFAR-10

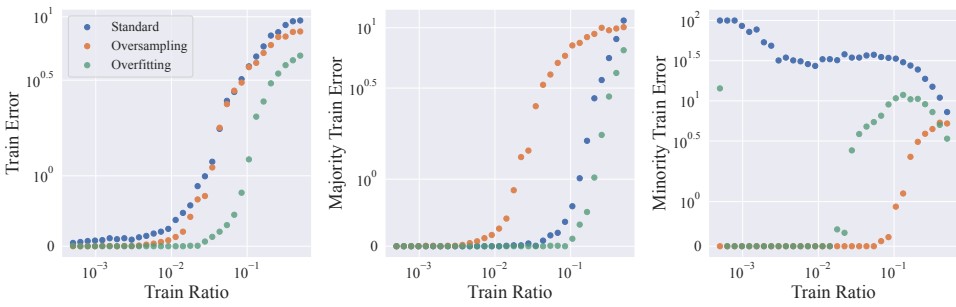

Figure 16: XGBoost train error on Adult

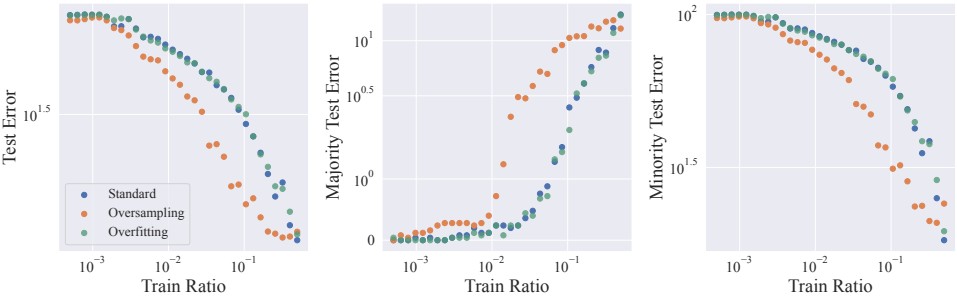

Figure 17: XGBoost test error on balanced test sets from Adult

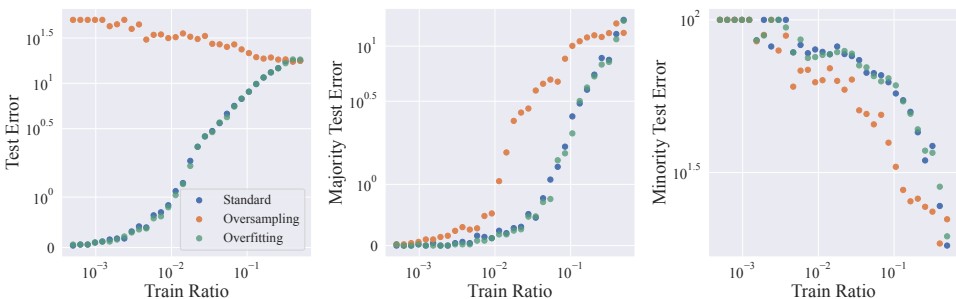

Figure 18: XGBoost test error on imbalanced test sets (train ratio and test ratio are equal) from Adult

