# OpenReview forum: "On Representation Learning Under Class Imbalance"
_ICLR.cc/2023/Conference — Submitted to ICLR 2023_

### Official Review · Reviewer_4VFj · 2022-10-21

**Confidence:** 3
**Correctness:** 2
**Technical Novelty And Significance:** 1
**Empirical Novelty And Significance:** 2
**Recommendation:** 3

**Clarity, Quality, Novelty And Reproducibility:**

The paper is generally well written.  However, the contribution is low due to a small number of datasets and only one performance metric.    Reproducibility seems to be ok.

**Strength And Weaknesses:**

Strengths:

1.  Instead of assuming the test set is balanced, they explore imbalance test ratio.

2.  the paper is generally well written

Weaknesses:

1.  More datasets would be more convincing--one image dataset and two tabular datasets seem insufficient

2.  accuracy/error is one measurement of performance, however, for imbalanced datasets, other performance measurement such as F1 and AUC of ROC could be used.

**Summary Of The Paper:**

The authors explore and evaluate several techniques for class imbalance.   Instead of assuming the test set is balanced, they explore imbalance test ratio.   Based one one image dataset with NN and two tabular datasets with SVM and XGBoost, they made the following observations.

1.   training ratio similar to test ratio is desirable
2.   oversampling minority samples does not improve generalization
3.   self-supervised learning for pre-training before supervised learning improves accuracy
4.   using Bayesian inference to incorporate uncertainty increases accuracy



**Summary Of The Review:**

The claims could be more convincing if more datasets and performance metrics (for imbalanced data) are used.  Currently, the claims seems to be stronger than the evidence can support.   The paper title stated representation learning, but some of the claims are not related to representation learning.

---

### Official Review · Reviewer_aUfE · 2022-10-23

**Confidence:** 5
**Correctness:** 3
**Technical Novelty And Significance:** 2
**Empirical Novelty And Significance:** 2
**Recommendation:** 3

**Clarity, Quality, Novelty And Reproducibility:**

## Clarity
Good.

Overall the writing is good. The structure is fine. Several typos exist.

## Quality
Fair.

Figures and tables are well presented. Several interesting visualizations.

## Novelty
Not good.

As I mentioned in the weakness part, almost all of the main observations in this paper have been extensively studied in the literature. I had a hard time figuring out what are the real contributions in the paper. Seems nothing new is really introduced here.

See the above weakness part for details.

## Reproducibility
N/A

No code is provided, though it seems straightforward to reproduce the results here, as many of them have been validated in the literature.

**Strength And Weaknesses:**

# Strength

+ The topic is interesting and timely. Data imbalance and few-shot learning is important and practical in real-world datasets, and in-depth analysis is plausible for understanding the behavior of representation learned under such data bias.

+ The writing and overall structure is good, and the paper is easy to follow.
---
# Weaknesses

Unfortunately, there are several major weaknesses that exist in the current paper.

### Unclear Contributions
The paper focuses on representation learning under class imbalance. However, many topics in this paper have already been extensively studied in the literature. I had a hard time figuring out the real contributions of this paper. Can you explain what's new in the paper, and position your work w.r.t the literature?

**Specifically**, the following "findings" in this paper have been well studied / known in the literature.
- _"re-balancing training data actually harms accuracy under imbalanced testing"_
1. This argument is actually trivial in its form. Without any label shift between training and testing, ERM should be the optimal solution [1]. This phenomenon actually motivates the logit adjustment method, where you only need to adjust the logit to compensate the training / testing label shift for a optimal classifier [1]. Going back to this paper, if the training / testing label distribution is the same, no matter how imbalanced they can be, you simply do not need to tune anything, as it is simply a in-domain generalization problem without any shifts.
2. Even under the above setup, the implicit assumption we have is $P_{train}(x|y) = P_{test}(x|y)$, where the only bias comes from the label shift between training and testing. Elaborating this assumption could potentially be a more interesting direction, which is however totally missing in the paper.
3. Perhaps the real interesting question here, is what if the test set is not balanced, but arbitrarily imbalanced (e.g., could be inversely long-tailed compared to the training label distribution). The observation here will easily break when extending to arbitrary testing label shifts. Such scenario is, however, also covered in the literature [2].

- _"oversampling which fit these samples do not improve generalization"_

1. Again, this is not even a proper takeway -- tons of papers in this field have already studied this phenomenon, which is why methods like logit adjustment, ensemble learning, self-supervised learning, contrastive learning, semi-supervised learning (with more unlabeled data) have been introduced. This observation is **out-dated** and do not provide anything new to the research field.

- _"A two-step process of (a) SSL pre-training on the imbalanced training set and ... accuracy across imbalance ratios."_

1. This takeaway is what has been exactly done in the paper two years ago [3]. Strictly speaking, nothing new is presented here, despite that less analysis is conducted in this paper. I suggest the authors doing a comprehensive literature review and correcly position the paper to the literature, and what new contribution/observation is made.

- _"Under class imbalance, models are particularly underspecified by the data"_

1. Similarly, past works (e.g., [4]) have studied this phenomenon. There are even new methods that involve uncertainty modeling for combating underspecification proposed [4] in imbalanced learning.

### Limited Experiments

- One big drawback in the experiments is that only small datasets are evaluated. CIFAR-10 has been standard but also too small for the data imbalance / long-tailed recognition field. The fields have advanced to larger and more practical datasets with higher resolution, such as ImageNet-LT or iNaturalist. Without validation on these large-scale datasets, the observations may not even be justifiable or convincing.

- No ablation studies across datasets / network architectures / optimization methods performed, which again makes the observation less convincing. If the author wanted to do an in-depth analysis, the first thing they need to make sure is that the observations **really persist** across a range of datasets / network architectures / etc.

- Despite that the title is "Representation learning ...", no real analysis on the learned feature space under class imbalance is performed. How exactly is the feature or representation balanced / imbalanced? More analysis and experiments are needed.

### Writing and typos
- page 4, "Takeaway" - the sentence is grammarly incorrect. You might want to change it to something as "Training with a train set exhibiting class imbalance similar to that of"

# References
[1] Long-tail learning via logit adjustment. ICLR 2021.

[2] Self-Supervised Aggregation of Diverse Experts for Test-Agnostic Long-Tailed Recognition. 2021.

[3] Rethinking the Value of Labels for Improving Class-Imbalanced Learning. NeurIPS 2020.

[4] Striking the Right Balance With Uncertainty. CVPR 2020.

**Summary Of The Paper:**

This paper studies representation learning under data imbalance. In particular, it performs extensive empirical investigation on the learning behaviors for different sets of models, and observe that  re-balancing class-imbalanced data is in general ineffective. Along this line, it studies self-supervised pre-training, Bayesian inference, and  flatness-seeking regularization for data imbalance, and verifies the advantages of these methods in the presence of imbalanced data across a variety of domains.

**Summary Of The Review:**

The topic of this paper is interesting. However, as elaborated in the weakness part, none of the observations in the paper is actually new. The contribution is unclear and the novelty is low.

Currently there are several major drawbacks and weaknesses, in terms of contribution and exepriments. I view this work as a half-baked work and needs significant improvements.

In summary, I recommend rejection for the current paper. The paper needs to be significantly improved to meet the bar of top conferences like ICLR.

---

> ### Author Response · Authors · 2022-11-17
> **Answer to Reviewer aUfE**
>
> Thank you for your extensive and encouraging feedback, and we are glad you find the topic interesting and timely. We included a general comment to all reviewers and ACs, which contains a description of our contributions, and we address your additional comments below:
>
> - __The contribution of the paper:__  We added the paper's contribution to the general response above.
> - __Position the work w.r.t the literature :__
>
>      - __"...ERM should be the optimal solution" -__  This statement is not true, nor is it supported by [Menon et al., 2021], and we see in our experiments that there are cases where the best performance is achieved not with the same training data ratio, and we should tune it. We agree that if we assume that we learned the  **true** p_train(x|y)  from our training data, and p_train(x|y)==p_test(x|y), namely, we have perfect generalization, then the logit adjustment method will achieve optimal results. However, in practice, we do not have (and we do not assume) either of these assumptions. Regarding your suggestion for testing on the arbitrarily imbalanced testing dataset, this is an interesting question that should be investigated. However, as per our knowledge, Zhang et al. (2021) did not study what the optimal training set ratio for different sets ratios is
>
>     - __"Oversampling which fits these samples does not improve generalization" -___  We are not claiming the oversampling methods achieve recent state-of-the-art results. As mentioned in the paper, we are aware that many recent methods have made performance improvements. However, to our knowledge, there has not been a method that directly studies the over-fitting of minority class samples.
>
>     - __"A two-step process of (a) SSL pre-training on the imbalanced training set has been done before"-__ As mentioned explicitly in our manuscript, our work on SSL was inspired by previous works that studied the effect of SSL pre-training for transfer learning in the context of class-imbalance [Liu et al., (2021), Zhang et al.,  (2021), Kotar et al., (2021)]. However, to our knowledge, Zhang et al.  (2021) did not use SSL pre-training and then fine-tuning on the same exact class-imbalanced dataset. To our knowledge, our study on SSL under class imbalance is novel and not present in the existing literature.
>
>
>     - __"Under class imbalance, models are particularly underspecified by the data" -__ We thank the reviewer for pointing out the work of Khan et al., (2019). In their paper, they use dropout to estimate the uncertainty of each class and reweight it according to the decision boundaries of the class. However, we suggested using regular on-the-shelf bayesian inference methods without special adjustment to imbalance learning.
>
> - __Limited Experiments:__ We thank the reviewer for this comment and agree that although the results look promising, validating them on a broader range of datasets/architectures will increase confidence in our conclusions. It is important to note that we tested three different datasets in two different domains (vision, tabular) using three different ML models (Neural networks, SVM, and gradient-based decision trees)
> - __No real analysis on the learned feature space under class imbalance is performed:__ We agree that a thorough study of intermediate features attained under class-imbalanced problems would be an interesting study for future work but is beyond the scope of our studies. Our focus is instead performance-driven, and we measure performance using a variety of metrics.
>
> **References**
>
> [Menon et al., 2021] Long-tail learning via logit adjustment \
> [Zhang et al., 2021] Self-Supervised Aggregation of Diverse Experts for Test-Agnostic Long-Tailed Recognition \
> [Liu et al., 2021] Self-supervised Learning is More Robust to Dataset Imbalance \
> [Kotar et al., 2021 Contrasting contrastive self-supervised representation learning pipelines\
> [Khan et al., 2019] Striking the Right Balance With Uncertainty

---

> > ### Comment · Reviewer_aUfE · 2022-12-11
> > **Response to Authors**
> >
> > First of all, I thank the authors for providing additional experiments and clarifications. I appreciate the efforts the authors made during the discussion phase.
> >
> > Unfortunately however, my concerns are not addressed. The work is not correctly positioned w.r.t the literature - I do not intend to repeat myself, but many of the concepts are indeed explored in the literature. Please refer to my original review for details.
> >
> > Further, making "Limited Experiments" / "analysis on the learned feature", etc. as futrue work is fine, but it significantly lower the contributions as well as insights of the paper.
> >
> > Considering the above as well as taking other reviewer's comments into account, I decide to keep my original rating. I do believe that some of the techniques and observations in the paper will have potential impacts for a broader audience in the field, but it will need extra careful revisions to incorporate the results and discussions with all reviewers, as well as more insights behind the observations.

---

### Official Review · Reviewer_D5Wc · 2022-10-24

**Confidence:** 3
**Correctness:** 2
**Technical Novelty And Significance:** 2
**Empirical Novelty And Significance:** 2
**Recommendation:** 5

**Clarity, Quality, Novelty And Reproducibility:**

Most of the parts of the paper are clearly written. Only a description of some experiments lacks details. In particular, in the case of the experiment in sec 4.1., it is unclear what it is "the best test ratio". It is also not clear if the number of examples is fixed when varying the class-imbalance ratio. In Figure 6, "SSL Train Data" and "SSL Linear Evaluation" probably refer to the proposed methods, but the names are hard to match with the methods in the text.

Evaluation of benefits of Bayesian learning/inference and Sharpness-Aware Minimization for representation learning in a class-imbalanced setting is novel. The first method, SSL pretraining followed by fine-tuning on the same class-imbalanced dataset, seems novel but incremental change.

For perfect reproducibility, the authors need to provide an exact description of the test protocol, e.g. what the number of the training and test examples was, or how the distribution in the intermediate classes was varied.

**Strength And Weaknesses:**

Strength:

The paper addresses a topic of both practically relevant and theoretical interest.

The authors identify three existing methods that show promising results when applied in class-imbalanced setting.

Weaknesses:

The authors make a relatively strong conclusion about previously used methods, e.g., that the rebalancing of training data is generally not helpful. The authors' conclusions regarding representation learning are made on the basis of experiments, which are reasonable; however, they use a single NN architecture trained on a single dataset. For me, it is not clear that the observations would apply to different data distributions. A similar problem, i.e. using a single dataset, appears when evaluating the three proposed methods.

Quantities used to characterize data distribution and performance have flows that limit conclusions made and should be at least discussed. First, the class-imbalance ratio used as a way to characterize the entire class distribution completely neglects the intermediate classes, which certainly have a large impact on the results. Second, the classification error is not an appropriate metric in the cases where the test data are severally class-imbalanced, as the performance on the minority class has a negligible influence on the metric.

For some unknown reason, the three proposed techniques are not compared against each other in terms of prediction performance nor in terms of computational demands.


**Summary Of The Paper:**

The paper addresses the learning of classifiers from class-imbalanced data. In the first part of the paper, the authors empirically investigate the effect of imbalanced classes on the performance of classification models. They find that i) oversampling minority class is ineffective, and that ii) adding too many examples of the majority class can be harmful as well. In the second part, they show that three existing methods can improve the performance of deep models in a class-imbalance scenario, namely i) using self-supervised pretraining followed by a supervised fine-tuning, ii) using Bayesian inference, and iii) using sharpness-aware minimization of the loss.

**Summary Of The Review:**

The paper comes with interesting observations regarding the usefulness of existing methods in a class-imbalanced setting. The main deficiency of this purely empirical paper is the limited number of datasets, which makes the conclusions rather speculative.

---

> ### Author Response · Authors · 2022-11-17
> **Answer to Reviewer  DW5c**
>
> Thank you for your extensive and encouraging feedback, and we are glad you find that our paper shows promising and novel results and addresses a topic of both practically relevant and theoretical interest.\
> We’ll address your comments below:
>
>
> - __Using more datasets for evaluation:__ We thank the reviewer for this comment and agree that although the results look promising, validating them on different data distributions will increase confidence in our conclusion. We would like to mention that we evaluated on three different datasets in different domains (vision and tabular) with different model types (Neural networks, SVM, and gradient-based decision trees).
>
> - __Using more performance metrics:__ As mentioned in the general response, we added several other metrics to our manuscript. These metrics include F1 score, AUC, precision, and recall. Regarding the class-imbalance ratio, this is a standard ratio used in previous works [Liu et al., 2021]. The intermediate classes are taken into account as we use exponential decay to determine their number. We agree that it is an interesting question to analyze the effect of the rate of how the intermediate classes decay and investigate different methods for determining the number of examples for the intermediate classes.
>
> - __Compared the three proposed techniques (computational demands) -__  We are currently comparing the proposed me techniques and will include it in our next updated draft.
> The description of some experiments lacks details. We thank the reviewer for this comment. The “best train ratio” is the ratio of the imbalance of the training dataset that achieved the best performance (the lowest error in Figure 1) on the given test ratio.  Indeed the number of examples is fixed when varying the class-imbalance ratio. Also, in Figure 6, "SSL Train Data" refers to the SSL pre-training on the same imbalance data, and "SSL Linear Evaluation" refers to SSL pre-training and then doing a linear evaluation on the top of the frozen features. We have now updated our manuscript to make these clarifications.
> An exact description of the test protocol:  For multiclass problems, we used the same method as in [Liu et al., 2021] to construct our imbalance dataset, namely first, we set the number of samples of the lowest class (based on the specific class ratio). For determining the number of intermediate classes, we used an exponential decay interpolation between the class with the most samples and the one with the fewest samples.
>
> **References**
>
> [Liu et al., 2021] Self-supervised Learning is More Robust to Dataset Imbalance

---

### Official Review · Reviewer_Mds7 · 2022-10-26

**Confidence:** 3
**Correctness:** 2
**Technical Novelty And Significance:** 2
**Empirical Novelty And Significance:** 2
**Recommendation:** 3

**Clarity, Quality, Novelty And Reproducibility:**

The novelty of this paper is not significant.

The source code is not provided.


**Strength And Weaknesses:**

Strengths

- From experiments, the authors find interesting observations for class-imbalance cases

Weakness

- Since this paper considers imbalanced data sets, it is not appropriate to use accuracy as the unique performance metric. Other metrics like Precision, recall, bias, f1, CSI, ROC, should be considered.

- The authors should discuss and implement recent works addressing the class-imbalance problem.

- Messages from this paper should be compared with recent works. For instance, one of the claims of this work is "minority samples are hard to fit, yet algorithms which fit them, such as oversampling, do not improve generalization." However, there are some works that improve the performance by fitting the minority.


**Summary Of The Paper:**

This paper considers the class-imbalance problem. In many practical cases, the data set has minority classes that have few samples. To understand the effect of highly imbalanced data sets, the authors perform extensive empirical studies for various ML models, including neural networks, gradient-boosted decision trees, and VMSs. The observations are as follows. Re-balancing and oversampling for the minor samples are not effective. They even degrade performance. Self-supervised pre-training is insensitive to imbalance. Bayesian is effective. Flatness-seeking regularization can provide more margin to the boundary of the minorities.


**Summary Of The Review:**

This paper provides many experimental results and messages from the experiments for the class-imbalance problem. However, this work does not compare with recent algorithms and the performance metric is not proper.

---

> ### Author Response · Authors · 2022-11-17
> **Answer to Reviewer Mds7**
>
> Thank you for your helpful feedback,  and we are glad you find our observation interesting. \
> We address each comment below:
>
> - __Using more performance metrics__ - As mentioned in the general response, we added several other metrics to our manuscript. The metrics included F1 score, AUC, precision, and recall. The metrics showed similar trends to the previous ones. It is important to emphasize that, in our initial manuscript, we also reported separate accuracy results for minority and majority classes. Unlike the accuracy of both classes combined, this measure is sensitive to the minority class performance regardless of how imbalanced the dataset is.
> - __Comparing to recent methods addressing class__- imbalance data. Our current study aims to gain new insights into learning under class-imbalanced data. We showed that even simple methods, which didn't design specifically for class-imbalance training and focused on learning better feature representation achieved good results on imbalanced datasets. We agree that analyzing recent methods and understanding their pipelines is an interesting direction, but it is outside this project's scope.
> - __Compare the message with recent works__ -  Some works improve the performance by fitting the minority - While some methods which fit the minority data may improve performance, what we uncover is that fitting the minority data itself does not improve performance, indicating that the success of methods which improve performance under class imbalance cannot be explained simply by fitting minority samples.

---

### Author Response · Authors · 2022-11-17
**General Response to Reviewers and AC**

We want to thank the reviewers for their thoughtful reviews. We are pleased that the reviewers recognize the potential and novelty of this manuscript and its concept. We here provide a general response addressed to all reviewers and ACs, as well as individual replies to address specific reviewer concerns as separate posts.

We would like to emphasize, as many of the reviewers have mentioned, the timeliness, novelty, and importance of this work, which tries to understand classification from highly imbalanced data sets. Using various ML models, including neural networks, gradient-boosted decision trees, and SVMs, we conducted an empirical study to explore the effects of class-imbalanced data on optimization and generalization. Our results indicate that differences between the train and the test data have a major effect on the learned representation and must be addressed. However, simple methods to address these differences, such as re-balancing and oversampling, are not effective and solve the wrong problem, namely, re-balancing the training data is not helpful, and fitting minority samples does not improve generalization.

Following these experiments, we analyze three factors that may affect the training of imbalanced datasets. Finally, we suggest three existing methods which improve the performance without ad-hoc interventions specific to imbalance training: (1) SSL, which is insensitive to imbalance and enchanted representation, (2) Bayesian inference, which deals with the underspecification problem in imbalance training and (3) sharpness-aware minimization which pulls decision boundaries away from minority samples. The success of these simple methods, which are not explicitly designed for imbalance training, emphasizes the importance of learning better feature representation and the need to focus on representation learning to address imbalance training.

To address the reviewers' concerns, we added additional performance metrics to the manuscript, which show the exact same trends as the metrics which were already in our draft. Following the reviewers' comments, we replicated our previous experiments and included the F1 score, AUC, precision, and recall metrics, and we added these experiments to Appendix B  of our updated draft. Notably, we observe the superior results achieved by the three simple methods we consider, SSL, bayesian inference, and SAM. We want to emphasize that in our initial manuscript, we also reported separately on the accuracy for the minority and majority classes. In contrast to the accuracy on both classes together, this measure is sensitive to the minority class performance even for extremely high imbalanced datasets.

We agree with the reviewers that although the results look promising, more datasets are required to fully support our findings and ensure they are generalized. Even though we showed that our results generalize to different domains and ML models, we agree that we should make sure that they persist across a range of datasets and models, and we would add them to the next updated draft.

---

### Decision · Program_Chairs · 2023-01-20

**Decision:**

Reject

**Justification For Why Not Higher Score:**

There was a general consensus that the findings of this paper are interesting, but there isn't sufficient analytical nor empirical support to entirely accept the claims or sufficiently evaluate the proposed solutions. Additionally more work is needed to make the paper more rigorous and contextualize with respect to recent work that also addresses some of these issues.

**Justification For Why Not Lower Score:**

N/A

**Metareview: Summary, Strengths And Weaknesses:**

The authors consider learning under class imbalance, challenging the validity of widely used techniques via empirical validation and proposing solutions to address problems encountered. Based on experiments with the CIFAR dataset for neural networks and and {Adult, Forest Cover} tabular datasets with SVM and XGBoost, they make the observations itemized below:
- It is desirable to have the training label distribution be similar to the test label distribution (i.e., under/over sampling can lead to poor generalization performance under the assumptions in this paper)
- Minority samples are difficult to fit, but oversampling minority samples does not necessarily improve generalization on test data.
- Self-supervised learning for pre-training on the imbalanced dataset followed by supervised fine-tuning improves accuracy consistently across different imbalance ratios.
- Bayesian inference consistently improves performance on imbalanced data by mitigating uncertainty within the estimator.
- Sharpness-aware minimization appears to improve performance by increasing the decision boundary margin relative to the minority examples.

The consensus strengths of this work include:
- Class imbalance is a widely-studied problem with significant practical implications in applied settings.
- The authors do make several interesting observations and provide methods to mitigate observed shortcomings; in particular, exploration of the train/test-imbalance ratios are interesting and seem a direction for needed further understanding.
- The paper is well-written and easy to understand.

The notable weaknesses of this work include:
- The exact contribution isn't clear; the 'controversial claims' with respect to existing methods aren't precisely stated enough (i.e., in mathematical form) to effectively evaluate. The reviewers pointed out several existing methods that deal with some interpretations of the the sampling-centric findings, which need to be addressed. Additionally from an empirical perspective, (1) claims that are counter to accepted understanding require evidence at the level of a systematic review and/or meta analysis or large-scale new experiments whereas this work has a limited number of datasets & experiments and (2) using accuracy as the metric for demonstrating this is clearly insufficient. While this second point was partially addressed during the rebuttal period, there wasn't sufficient interpretation/analysis to convincingly support these statements.
- There also isn't any theoretical analysis to support these claims (which would be needed given the lack of large-scale experiments). Additionally, there is a general absence of mathematical rigor to precisely understand what methodological shortcomings are being addressed in each case (as the improvements may not be due to the stated reasons). This could also be supported with ablation studies to some degree.
- Regarding the SSL, Bayesian, and SAM findings, there are existing works that require additional discussion and in some cases, direct comparison. The reviewers felts overall that there wasn't sufficient contextualization and contrastive evaluation wrt existing relevant work.

Overall, the consensus was that there are some potentially interesting findings. However, the work needs more contextualization/comparison wrt recent existing work, more experiments to adequately support the paper's claims, and more general rigor with respect to the core claims. Some interesting directions, but just needs more work to be a strong contribution to this subfield.